# Increased Antibiotic Susceptibility of Gram-Positive Bacteria in Cerebrospinal Fluid Compared to Broth

**DOI:** 10.3390/antibiotics13121215

**Published:** 2024-12-14

**Authors:** Jennifer S. Wirth, Marija Djukic, Katrin Biesner, Utz Reichard, Roland Nau, Jana Seele

**Affiliations:** 1Department of Geriatrics, Evangelisches Krankenhaus Göttingen-Weende, 37075 Göttingen, Germany; j.wirth@stud.uni-goettingen.de (J.S.W.); mdjukic@gwdg.de (M.D.); katrin.biesner@stud.uni-goettingen.de (K.B.); jana_seele@gmx.de (J.S.); 2Department of Neuropathology, University Medical Center Göttingen, 37073 Göttingen, Germany; 3MVZ Wagnerstibbe, Amedes Group, 37081 Göttingen, Germany; ureicha@gwdg.de

**Keywords:** *Enterococcus faecalis*, *Staphylococcus aureus*, *Staphylococcus epidermidis*, *Streptococcus pneumoniae*, minimal inhibitory concentration, minimal bactericidal concentration, cerebrospinal fluid

## Abstract

**Background:** In hospital- and community-acquired central nervous system infections, resistant Gram-positive bacteria are an increasing therapeutic challenge. The present approach does not attempt to identify rapidly bactericidal therapies for susceptible pathogens but aims to improve methods to find antibiotic regimens for multi-resistant pathogens that are effective in vivo in spite of reduced in vitro susceptibility in culture media. **Methods:** Antibiotic susceptibility was tested in cerebrospinal fluid (CSF) and Mueller–Hinton broth (*Enterococcus faecalis*, methicillin-resistant *Staphylococcus aureus*, *Staphylococcus epidermidis*) or brain–heart infusion (*Streptococcus pneumoniae*). **Results:** Minimal inhibitory concentrations (MICs) and minimal bactericidal concentrations (MBCs) were either lower in CSF than in broth or equal in CSF and broth. The difference between MICs in CSF and broth was prominent with gentamicin, levofloxacin, linezolid (staphylococci), and vancomycin (staphylococci and pneumococcus), whereas it was absent with ampicillin (*E. faecalis*), penicillin G (*S. pneumoniae*), linezolid (enterococcus and pneumococcus), and vancomycin (enterococcus). In no case was the MIC or MBC higher in CSF than in broth. **Conclusions:** Several antibiotics possess an antibacterial effect in CSF at lower concentrations than the MICs determined in broth, i.e., MICs in broth underestimate in situ susceptibility in CSF.

## 1. Introduction

As a consequence of the presence of the blood–brain and blood–cerebrospinal fluid (CSF) barriers, the concentrations of many antibiotics in the central nervous compartments are far below those in plasma. Moreover, the central nervous system (CNS) cannot be considered a single homogeneous compartment [1,2]. Therefore, the treatment of CNS infections is a therapeutic challenge [3,4,5,6]. Previous data suggest that antibiotic CSF concentrations of approx. 10 times above the minimal inhibitory concentration (MIC) determined in culture media are necessary for optimum therapies for CNS infections [3,4,5].

In most countries, the microorganisms causing community-acquired bacterial meningitis are fortunately still susceptible to established antibiotics, but the MICs of these antibiotics are gradually rising [7,8]. Conversely, for hospital- and community-acquired CNS infections caused by carbapenem- and colistin-resistant Gram-negative, vancomycin-resistant Gram-positive, and multi-resistant pneumococcal strains, therapeutic options are scarce [9,10,11]. For multi-resistant pathogens, CSF concentrations 10× ≥ MIC often cannot be achieved by intravenous antibiotic therapy because of toxic adverse effects.

To predict the in vivo efficacy of antimicrobials against bacterial strains isolated from humans, generally, the MIC is determined in vitro either by the macrodilution or microdilution method or by the E-test. Most frequently, the qualitative disc diffusion test is used to assess antibiotic susceptibility. Compared to more time-consuming in vitro and in vivo tests, the E-test and the disc diffusion test are easy to handle and comparatively cheap. These tests are usually conducted in standard media such as Mueller–Hinton broth (MHB) or on standard plates such as sheep blood agar. Culture media are ideal for bacterial growth, but their composition does not resemble the conditions in body fluids [12,13]. Different nutrient compositions, the lack of immune cells, physiological proteins, and host–bacterial interactions can lead to a misestimation of in vivo susceptibility when determining MICs in culture media [14]. The present approach does not attempt to identify rapidly bactericidal therapies for susceptible pathogens but aims to improve methods to find antibiotic regimes for multi-resistant pathogens that are effective in vivo in spite of reduced in vitro susceptibility of the pathogens in culture media.

## 2. Results

Bacterial growth was monitored prior to the determination of MICs and MBCs, and curves revealed that all bacteria studied grew well in MHB (BHI for *S. pneumoniae*). Bacterial growth in CSF was slower than in MHB/BHI, but sufficient for MIC determination for all strains studied (Figure 1A). A prolongation of the incubation time for the determination of the MIC/MBC from 24 h to 48 h did not result in substantial differences in the MICs/MBCs measured.

The *S. pneumoniae* strain studied showed a reduced sensitivity to penicillin G; according to EUCAST criteria (Table 1), penicillin G is not suitable for pneumococcal meningitis caused by a strain with a MIC ≥ 0.06 mg/L [15]. The other bacteria studied had MICs below or equal to the MIC considered susceptible by EUCAST [15]. We selected strains with a MIC close to the cut-off according to EUCAST (Table 1). Since most EUCAST breakpoints were not designed for bacterial meningitis (exception: *S. pneumoniae*), low antibiotic concentrations in the central nervous compartments can cause treatment failures in CNS infections by organisms with MICs close to the breakpoints [6]. The *E. faecalis* strain studied was resistant to gentamicin used as monotherapy (no EUCAST breakpoint available). Its MIC indicated that synergy with penicillins or glycopeptides could be expected [15]. MICs and MBCs were either lower in CSF than in broth or equal in CSF and broth. In no case was the MIC or MBC higher in CSF than in broth. Details can be found in Table 1. The difference between MICs in broth and CSF was prominent (median MICs [mg/L]; *p* < 0.05) with vancomycin (*S. aureus*: 1 vs. 0.25; *S. epidermidis*: 2 vs. 0.5; *S. pneumoniae*: 1.5 vs. 0.125), gentamicin (*S. aureus*: >16 vs. 8; *E. faecalis*: 16 vs. 4), levofloxacin (*S. aureus*: 1 vs. 0.1875; *S. epidermidis*: 1 vs. 0.1875; *S. pneumoniae*: 1 vs. 0.125), linezolid (*S. aureus*: 2 vs. 0.75; *S. epidermidis*: 2 vs. 0.5), and cefuroxime (*S. epidermidis*: 1 vs. 0.5) (Figure 1B–F), whereas it was absent with ampicillin (*E. faecalis*), penicillin G (*S. pneumoniae*), linezolid (enterococcus and pneumococcus), and vancomycin (enterococcus). With respect to the activity of linezolid against *S. pneumoniae*, only the MBC was significantly lower in CSF than in BHI (median MBC 1 vs. 2 mg/L), whereas the MBCs of levofloxacin against *S. pneumoniae* in CSF and BHI were approx. equal. The difference between MICs and MBCs was most prominent with vancomycin for *S. aureus* in MHB (1 vs. 16 mg/L). For the other antibiotics and bacteria, MICs and MBCs were equal or maximally one concentration higher than the MIC in MHB.

**Figure 1 antibiotics-13-01215-f001:**
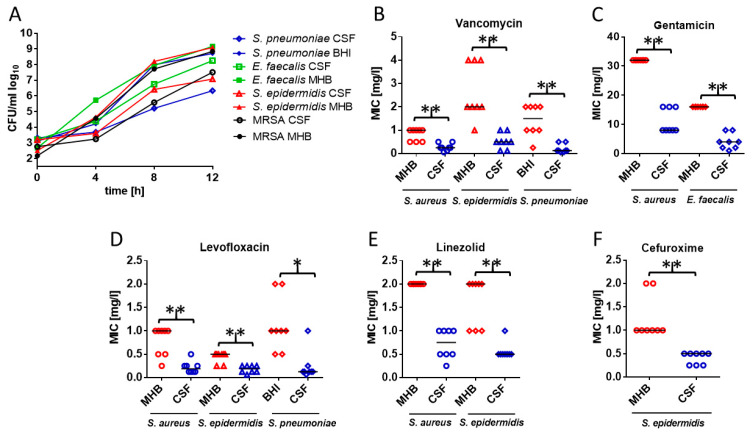
Growth curves (**A**) and minimal inhibitory concentrations (MICs) of *Staphylococcus aureus* ATCC 43300, *Staphylococcus epidermidis* ATCC 12228, *Enterococcus faecalis* ATCC 29212, and *Streptococcus pneumoniae* ATCC 43300 for different antibiotics in broth and cerebrospinal fluid (CSF). (**A**) All bacteria readily grew in CSF, allowing for the determination of MICs and MBCs in CSF without the addition of culture media. (**B**–**F**) MICs were determined 8 times on different days. Since CSF has a pH of approx. 10, when equilibrated with room air at 37 °C in the absence of CO_2_ [16], all MIC determinations were performed in the presence of 5% CO_2_. Each symbol represents an individual measurement; the horizontal bars represent the medians. Only bacteria and antibiotics with substantial differences in susceptibility in broth and CSF are depicted (* *p* < 0.05; ** *p* < 0.01 Wilcoxon matched-pairs signed rank test).

**Table 1 antibiotics-13-01215-t001:** Minimal inhibitory concentrations (MICs) and minimal bactericidal concentrations (MBCs) of *Enterococcus faecalis* ATCC 29212, methicillin-resistant *Staphylococcus aureus* ATCC 43300, *Staphylococcus epidermidis* ATCC 12228, and *Streptococcus pneumoniae* ATCC 43300 in cerebrospinal fluid (CSF) compared to Mueller–Hinton both (MHB) */brain–heart infusion (BHI) ^#^. Median (range) of 8 independent measurements in broth and different CSF samples.

Strain	Antibiotic	Resistance ^%^	MIC Broth [mg/L] Median (Range)	MIC CSF [mg/L] Median (Range)	MBC Broth [mg/L] Median (Range)	MBC CSF [mg/L] Median (Range)	*p* MIC	*p* MBC
*E. faecalis*	Ampicillin	>8	2 (1–4)	2 (0.5–8)	4 (1–8)	4 (2–8)	0.80	0.81
	Gentamicin	^$^	16 (16–16)	4 (1–8)	16 (16–16)	12 (4–16)	0.008	0.13
	Linezolid	>4	2 (2–2)	0.75 (0.5–4)	Nd ^+^	Nd ^+^	0.80	Nd ^+^
	Vancomycin	>4	2 (1–4)	1.5 (0.5–2)	4 (2–8)	4 (2–8)	0.13	0.19
*S. aureus*	Gentamicin	>2	>16 (>16–>16)	8 (8–16)	>16 (>16–>16)	16 (8–16)	0.008	0.008
	Levofloxacin	>1	1 (0.25–1)	0.1875 (0.125–0.5)	1 (0.25–2)	0.25 (0.06–0.5)	0.008	0.008
	Linezolid	>4	2 (2–2)	0.75 (0.25–1)	Nd ^+^	Nd ^+^	0.008	Nd ^+^
	Vancomycin	>2	1 (0.5–1)	0.25 (0.06–0.5)	16 (0.5–16)	0.25 (0.25–1)	0.008	0.016
*S. epidermidis*	Cefuroxime	>4	1 (1–2)	0.5 (0.25–0.5)	1 (1–2)	1 (0.25–2)	0.008	0.880
	Levofloxacin	>1	0.5 (0.25–0.5)	0.1875 (0.125–0.25)	0.5 (0.25–0.5)	0.25 (0.125–0.25)	0.008	0.016
	Linezolid	>4	2 (1–2)	0.5 (0.5–0.5)	Nd ^+^	Nd ^+^	0.008	Nd ^+^
	Vancomycin	>4	2 (2–4)	0.5 (0.125–1)	2 (2–16)	1 (0.125–4)	0.008	0.016
*S. pneumoniae*	Penicillin G	>0.06 ^&^	0.5 (0.25–1)	0.25 (0.06–0.5)	0.5 (0.25–1)	0.75 (0.25–1)	0.055	0.63
	Levofloxacin	>2	1 (0.5–2)	0.125 (0.06–1)	1 (0.5–2)	1 (0.5–2)	0.016	1.0
	Linezolid	>2	1 (1–2)	0.25 (0.125–2)	2 (2–2)	1 (0.25–2)	0.16	0.031
	Vancomycin	>2	1.5 (0.25–2)	0.125 (0.06–0.5)	1.5 (0.25–8)	0.375 (0.125–0.5)	0.008	0.016

*p*-value: Wilcoxon matched-pairs signed rank test. Nd: not determined. ^%^ breakpoints according to EUCAST criteria [15]. * *Enterococcus faecalis* ATCC 29212, methicillin-resistant *Staphylococcus aureus* ATCC 43300, and *Staphylococcus epidermidis* ATCC 12228. ^#^
*Streptococcus pneumoniae* ATCC 43300. ^+^ Linezolid MBC was not determined since time–kill studies demonstrated a bacteriostatic effect of linezolid against staphylococci and enterococci but a bactericidal effect against *S. pneumoniae* [17]. ^&^ Breakpoint for the indication of meningitis [15]. ^$^ Enterococci are considered resistant to gentamicin when used in monotherapy; no EUCAST breakpoint available. When MIC < 128 mg/L, the strain is considered wild-type, i.e., synergism with penicillins or glycopeptides can be expected.

## 3. Discussion

The current clinical practice of assessing the efficacy of antibiotics needs significant adjustments in how antimicrobial susceptibility testing is conducted, how infections are risk-stratified, and how antibiotics are utilized. This will require the (a) assessment of the action of antibiotics in physiologic media, (b) consideration of the host immune response in the evaluation of the action of an antibiotic, and (c) understanding of the influence of antibiotics on bacterial virulence [12,13,16,18]. Antibiotic susceptibility testing in media mimicking host environments succeeded in identifying antibiotics that were effective in bacterial clearance and host survival in vivo, although the same antibiotics failed when standard antimicrobial susceptibility testing (usually in MHB, unless specified otherwise, when the bacteria tested did not grow in MHB) was performed. Moreover, antibiotics were identified that were ineffective in vivo despite appearing to be effective in standard antimicrobial susceptibility testing [13].

Compared to broth designed for bacterial growth, CSF is a nutrient-deficient medium. Growth was slower in CSF than in MHB or BHI (Figure 1A). Bacteria may be more susceptible to antibiotics in nutrient deficiency [12,13]. As in the present study, in pooled CSF from patients in the presence of CO_2_, cefepime was bactericidal at a lower concentration (lowest concentration, 0.5× MIC) than in MHB (lowest concentrations, 2× MIC). In contrast, rifampicin was bactericidal at a lower concentration in MHB (lowest concentration, 1× MIC) than in CSF in the presence of CO_2_ (lowest concentration, 2× MIC). Because CSF has a pH far above 7.4 in the presence of room air without 5% CO_2_ supplementation, cefepime and rifampicin were the least effective in CSF without CO_2_ [19]. This illustrates that—as in the present study—supplementation with 5% CO_2_ is mandatory for the determination of clinically relevant MICs and MBCs in CSF. Similarly, against *S. aureus* (ATCC 29213) and *S. epidermidis* (ATCC 12228), lower linezolid concentrations in CSF were needed (1× MIC and 0.5× MIC) to achieve a bacteriostatic effect than in MHB (4× MIC for both strains) in the presence of 5% CO_2_ [20].

For fosfomycin, a concentration of 8× MIC was required in CSF despite the presence of 5% CO_2_ to achieve sustained bacterial killing compared to MHB supplemented with glucose-6-phosphate (1× MIC) [21]. This finding is probably explained by the different glucose-6-phosphate concentrations in CSF (approx. 1 mg/L) (5) compared to the 25 mg/L usually added as a supplement to MHB for the determination of fosfomycin MICs. In CSF without CO_2_, fosfomycin had no relevant antimicrobial effect even at 8× MIC [21].

As in previous observations with cefepime and linezolid, here, the MICs and MBCs determined in CSF in the presence of 5% CO_2_ to ensure a physiological pH were either lower or equal to those measured in broth. In no case was the MIC or MBC in CSF higher than in broth. One limitation of our study is that we used cell-free normal CSF or CSF with mild to moderate impairment of the blood–CSF barrier. In CSF from patients with meningeal inflammation, results of bacterial susceptibility testing may be slightly different for the following reasons: 1. CSF in meningeal inflammation often contains higher protein concentrations than the CSF used by us, which may facilitate bacterial growth. 2. Meningeal inflammation often causes lactate acidosis in CSF. Low pH diminishes the activity of several antibiotics. 3. CSF in meningeal inflammation contains higher CSF leukocyte counts. Leukocytes may phagocytose bacteria and thereby increase the antibacterial activity of antibiotics. Conversely, substrates from decayed leukocytes and erythrocytes may stimulate bacterial growth. The main limitation of our study, however, is the low number of bacterial species and strains tested. Because of the lack of CSF, we were unable to test more antibiotics or bacterial strains. The possible differences in MICs in CSF and broth will have to be studied in more bacterial strains and for more antibiotics in order to broaden our therapeutic options for CNS infections caused by multi-resistant bacteria. To collect enough CSF from patients for MIC/MBC studies is a great challenge in clinical routines. The collection of sufficient amounts of CSF from patients with meningeal inflammation not treated by antibiotics is impossible for ethical reasons.

In conclusion, in the present study, bacteria in CSF were either as susceptible as or more susceptible to antibiotics than in broth. This implies that some antibiotics inhibit the growth of bacteria in CSF at lower antibiotic concentrations than the MICs determined in standard broth. For this reason, in CNS infections where standard antimicrobial susceptibility testing does not yield adequate therapeutic options, it may be prudent to test the antimicrobial susceptibility of the causative bacterium in CSF.

## 4. Materials and Methods

### 4.1. CSF

CSF was collected from patients with suspected normal pressure hydrocephalus (n = 25). As part of the diagnostic routine, 40 mL of CSF was drawn by lumbar puncture. The CSF not necessary for routine laboratory analyses (approx. 30 mL) was used for this study. The CSF was not pooled. We deliberately did not pool CSF in order to detect and exclude CSF samples that yielded implausible results. All patients had normal CSF leukocyte counts (0–1/μL). Albumin, IgG, IgA, and IgM analyses in CSF and serum did not suggest CNS inflammation. Mild to moderate impairment of the blood–CSF barrier was present in 8 patients [maximum CSF protein content 870 mg/L (normal ≤ 450 mg/L), maximum CSF/serum albumin ratio 17.2 × 10^−3^ (normal in a 75-year-old person ≤ 9 × 10^−3^)]. Since proteins promote the growth of bacteria, an impairment of the blood–CSF barrier should facilitate bacterial growth in CSF. No patient received antibiotics in the two days before or on the day of the lumbar puncture. One CSF sample had to be excluded because antibiotic therapy until the day before the spinal tap initially had been overlooked.

### 4.2. Bacterial Culture and Determination of MICs and Minimum Bactericidal Concentrations (MBCs)

Bacteria preserved in glycerol cryostocks frozen at −80 °C were grown overnight in cation-adjusted MHB (Merck Millipore, Darmstadt, Germany) (*Enterococcus faecalis* ATCC 29212, methicillin-resistant *Staphylococcus aureus* ATCC 43300, *Staphylococcus epidermidis* ATCC 12228) or brain–heart infusion (BHI, bioMérieux, Nürtingen, Germany) (*Streptococcus pneumoniae* ATCC 49619). For security reasons, we tested bacterial strains with a reduced susceptibility to individual antibiotics, but we did not test multi-drug resistant bacteria. For the determination of MICs and MBCs, bacteria were diluted to a final concentration of 5 × 10^5^ CFU/mL in all incubations, either in MHB (*E. faecalis*, *S. aureus*, *S. epidermidis*) or in BHI (*S. pneumoniae*). CSF had a pH of approx. 10 when equilibrated with room air at 37 °C in the absence of CO_2_ [16]. This pH affected the survival of several bacterial species [16]. Therefore, all MIC determinations were performed in the presence of 5% CO_2_, which resulted in a pH of approx. 7.5, close to the physiological pH in the absence of meningeal inflammation [16]. Bacteria were incubated with the respective antibiotic at a final volume of 200 μL in 96-well microtiter plates (Sarstedt, Nümbrecht, Deutschland). The MIC was defined as the lowest antibiotic concentration where, after 24 h, no turbidity was visible. The strains used were provided by the American Type Culture Collection (ATCC). They were used as standard test strains in the MVZ Wagnerstibbe, Göttingen. In this laboratory for clinical microbiology, the MICs are determined by the Vitek^R^ 2 Test (Biomerieux, Nürtingen, Germany), an automated microbiology system utilizing growth-based technology. The MICs determined by us by broth micodilution in MHB or BHI were equal or a maximum of one titer step higher or lower compared to the MICs measured by the Vitek^R^ 2 Test. The MBC was defined as the lowest antibiotic concentration that caused a reduction of the bacterial concentrations by 3 orders of magnitude within 24 h, i.e., below 5 × 10^2^ CFU/mL. The MBC was measured after determination of the MIC by quantitative plating of 1:10 dilutions on sheep agar plates using the following concentrations of antibiotics: 1× MIC, 2× MIC, 4× MIC, 8× MIC. The following antibiotics were studied: ampicillin (treatment of choice for *E. faecalis* CNS infections), gentamicin (enhances the bactericidal effect of β-lactam antibiotics, suitable for intrathecal therapy), linezolid (treatment of choice for CNS infections by ampicillin-resistant enterococci, methicillin-resistant staphylococci, and cephalosporin-resistant *S. pneumoniae*), vancomycin (treatment of choice for CNS infections by ampicillin-resistant enterococci, methicillin-resistant staphylococci, and cephalosporin-resistant *S. pneumoniae*, suitable for intrathecal therapy), cefuroxime (reserves cephalosporin in CNS infections), and levofloxacin (broad spectrum, has excellent penetration into CSF) [6]. Linezolid MBCs were not determined for staphylococci and enterococci since time–kill studies demonstrated a bacteriostatic effect of linezolid for these pathogens [17].

### 4.3. Statistics

Statistical comparisons of MICs and MBCs in broth and CSF were compared by Wilcoxon matched-pairs signed rank tests using GraphPad Prism 6 software (GraphPad, San Diego, California, USA. An α error *p* < 0.05 was considered statistically significant.

### 4.4. Ethics

This study was approved by the Ethics Committee of the University of Medicine, Göttingen (approval reference number 31/2/2020). Patients gave their written informed consent to provide the CSF not necessary for routine laboratory analyses for this study.

## 5. Conclusions

In the present study, bacteria in CSF were either as susceptible or more susceptible to antibiotics than in broth. This implies that some antibiotics inhibit the growth of bacteria in CSF at lower antibiotic concentrations than the MICs determined in standard broth. In CNS infections, when standard antimicrobial susceptibility testing does not yield adequate therapeutic options, it may be prudent to test the antimicrobial susceptibility of the causative bacterium in CSF.

## Data Availability

The data presented in this study are available upon request from the corresponding author due to ethical reasons.

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
