# Peer review of "Increased Antibiotic Susceptibility of Gram-Positive Bacteria in Cerebrospinal Fluid Compared to Broth"

_antibiotics, 2024, doi:10.3390/antibiotics13121215_

Round 1
Reviewer 1 Report
Comments and Suggestions for Authors
The study is interesting; if the results obtained in clinical isolates of these bacteria are reproducible, it could become an alternative worth evaluating in the clinical microbiology laboratory to optimize the treatment of CNS infections.
Below are some observations regarding form:
- The terms in vivo and in vitro should be italicized throughout the text. Furthermore, they should not include hyphens in any case.
- Figure 1 is unclear. The details are not easily visible; it is suggested to either enlarge the size of the figures or use colors.
- In line 50, it is stated that the disk diffusion test is one of the methods to determine the MIC. This assertion is incorrect; the test is qualitative and does not determine the MIC. The paragraph needs to be corrected.
- In the methodology section, it would be helpful to provide more details about the methodology used to determine the MBC. The employed methodology is not clearly explained.
- To validate the MIC testing procedures, it is necessary to perform quality control using standard strains. Were quality controls conducted?
- In the discussion section, it would be beneficial to mention the study’s limitations.
Author Response
General suggestions
We described our quality control measures in the Methods section and presented the MICs measured in relation to the EUCAST breakpoints in the Results section and Table 1.
Specific suggestions
- The terms in vivo and in vitro should be italicized throughout the text. Furthermore, they should not include hyphens in any case.
We italicized in vivo and in vitro throughout the text. We deleted the hyphens. - Figure 1 is unclear. The details are not easily visible; it is suggested to either enlarge the size of the figures or use colors.
For the revised Figure 1 we used colors and enlarged the size of the writing. We will be happy, when the production team decides to enlarge the size of the figures in order to increase the visibility. - In line 50, it is stated that the disk diffusion test is one of the methods to determine the MIC. This assertion is incorrect; the test is qualitative and does not determine the MIC. The paragraph needs to be corrected.
We inserted the following sentence: „Most frequently, the qualitative disc diffusion tests is used to assess antibiotic susceptibility“. The next sentence was modified accordingly: „Compared to more time-consuming in -vitro and in -vivo tests, the E-test and the disc diffusion test are easy to handle and comparatively cheap“. - In the methodology section, it would be helpful to provide more details about the methodology used to determine the MBC. The employed methodology is not clearly explained.
We clarified: „The MBC was measured after determination of the MIC by quantitative plating of 1:10 dilutions on sheep agar plates using the following concentrations of antibiotics: MIC, 2 x MIC, 4 x MIC, 8 x MIC“. - To validate the MIC testing procedures, it is necessary to perform quality control using standard strains. Were quality controls conducted?
The strains used (Enterococcus faecalis ATCC 29212, methicillin-resistant Staphylococcus aureus ATCC 43300, Staphylococcus epidermidis ATCC 12228, Streptococcus pneumoniae ATCC 49619) were provided by the American Type Culture Collection (ATCC). They were used as standard test strains in the MVZ Wagnerstibbe, Amedes Group, Göttingen, Germany. In this laboratory for clinical microbiology, the MICs are determined by the VitekR 2 Test (Biomerieux, Nürtingen, Germany), an automated microbiology system utilizing growth-based technology. The MICs determined by us in MHB or BHI were equal or at a maximum one titer step higher or lower compared to the MICs measured by the the VitekR 2 Test. This was clarified in the Methods section. - In the discussion section, it would be beneficial to mention the study’s limitations.
We inserted the following sentences into the Discussion: „One limitation of our study is that we used cell-free normal CSF or CSF with mild to moderate impairment of the blood-CSF barrier. In CSF from patients with meningeal inflammation, results of bacterial susceptibility testing may be slightly different for the following reasons: 1. CSF in meningeal inflammation often contains higher protein concentrations than the CSF used by us, which may facilitate bacterial growth. 2. Meningeal inflammation often causes lactate acidosis in CSF. Low pH diminishes the activity of several antibiotics. 3. CSF in meningeal inflammation contains higher CSF leukocyte counts. Live leukocytes may phagocytose bacteria and thereby may increase the antibacterial activity of antibiotics. Conversely, substrates from decayed leukocytes and erythrocytes may stimulate bacterial growth. The main limitation of our study, however, is the low number of bacterial species and strains tested…. To collect enough CSF from patients for MIC/MBC studies is a great challenge in clinical routine. The collection of sufficient amounts of CSF from patients with meningeal inflammation not treated by antibiotics is impossible for ethical reasons.“

Reviewer 2 Report
Comments and Suggestions for Authors
Dear authors,
congratulations for your work.
I have some concernes, though, which i shall explain:
Abstract section: the pathogens you examined are not just hospital acquired but also community acquired pathogens for CNS infections.
Also, as you explained in your result section, the pathogens you examined were not multi-drug resistant. In fact, there were some susceptible ones.
I would suggest changing the purpose of the paper to include both community and hospital acquired pathogens, which may be multi-resistant.
Line 12: In “hospital acquired” rather than “nosocomial”
Line 42-43, 48-51: needs grammar check
Line 69-70: “Since bacterial growth in pure CSF was sufficient for MIC determination (Figure 67 1A), and the MICs/MBCs in the cell culture media listed above did not closely resemble 68 those in CSF, MICs/MBCs in DMEM and RPMI without and with supplementation by MHB and in CSF with supplementation by MHB were not further analyzed”à Maybe you could refine the sentence to make more sense. I would suggest simpler sentences.
In the results section, I would suggest mentioning the interpretation of susceptibility results. It is very important if a susceptible pathogen to an antimicrobial agent reports as resistant or remains susceptible and vice versa. Of concern, you could add a paragraph with the estimated major and very major errors.
Line 131: bibliography?
Thank you.
Comments on the Quality of English Language
No further comments.
Author Response
General suggestions
In the Introduction, we clarified the difference between quantitative and qualitative methods to assess susceptibility to antibiotics. We carefully revised the text for spelling and grammar mistakes.
Specific suggestions
- Abstract section: the pathogens you examined are not just hospital acquired but also community acquired pathogens for CNS infections.
We thank for this hint and included „and community-acquired“ in the Abstract section. - Also, as you explained in your result section, the pathogens you examined were not multi-drug resistant. In fact, there were some susceptible ones.
We clarified in the Methods section: „For security reasons, we tested bacterial strains with a reduced susceptibility to individual antibiotics, but we did not test multi-drug resistant bacteria.“ The reason is that at our institution we have very few susceptibility problems in CNS infections and that we do not possess a security lab suitable to work with multi-drug resistant pathogens imported from other institutions. - I would suggest changing the purpose of the paper to include both community and hospital acquired pathogens, which may be multi-resistant.
As suggested, we changed the purpose of the paper to include both community and hospital acquired pathogens, which may be multi-resistant. - Line 12: In “hospital acquired” rather than “nosocomial”
“nosocomial” was changed into “hospital-acquired”. - Line 42-43, 48-51: needs grammar check
We corrected the grammar of both sentences. - Line 69-70: “Since bacterial growth in pure CSF was sufficient for MIC determination (Figure 67 1A), and the MICs/MBCs in the cell culture media listed above did not closely resemble 68 those in CSF, MICs/MBCs in DMEM and RPMI without and with supplementation by MHB and in CSF with supplementation by MHB were not further analyzed”. Maybe you could refine the sentence to make more sense. I would suggest simpler sentences.
We deleted this sentence describing preliminary experiments in order not to cause confusion. - In the results section, I would suggest mentioning the interpretation of susceptibility results. It is very important if a susceptible pathogen to an antimicrobial agent reports as resistant or remains susceptible and vice versa. Of concern, you could add a paragraph with the estimated major and very major errors.
We included the criteria and data of susceptibility and resistance according to EUCAST into Table 1 and the Results section. An appropriate new reference [15] was listed in the References section [EUCAST Clinical Breakpoint Tables v. 14.0, valid from 2024-01-01. https://www.eucast.org/clinical_breakpoints (retrieved November 29, 2024)]. The following paragraph was inserted into the Results section: The pneumoniae strain studied showed a reduced sensitivity to penicillin G; according to EUCAST criteria (Table 1), penicillin G is not suitable for pneumococcal meningitis caused by strains with a MIC ≥0.06mg/L. The other bacteria studied had MICs below or equal to the MIC considered susceptible by EUCAST. We selected strains with a MIC close to the cut-off according to EUCAST (Table 1). Since most EUCAST breakpoints were not designed for bacterial meningitis (exception: S. pneumoniae), low antibiotic concentrations in the central nervous compartments can cause treatment failures in CNS infections by organisms with MICs close to the breakpoints [6]. The E. faecalis strain studied was resistant to gentamicin used as monotherapy (no EUCAST breakpoint available). Its MIC indicated that synergy with penicillins or glycopeptides could be expected. - Line 131: bibliography?
We inserted References 12 & 13 to substantiate our statement.

Round 2
Reviewer 2 Report
Comments and Suggestions for Authors
Dear authors,
thank you for considering my suggestions.
I wish you the best of luck.